# Emotional Intelligence, Interpersonal Relationships and the Role of Gender in Student Athletes

**DOI:** 10.3390/ijerph19159212

**Published:** 2022-07-28

**Authors:** Isabel Mercader-Rubio, Nieves Gutiérrez Ángel, Nieves Fátima Oropesa Ruiz, Pilar Sánchez-López

**Affiliations:** Department of Psychology, Faculty of Education Sciences, Universidad de Almería, 04120 Almería, Spain; imercade@ual.es (I.M.-R.); psanchez@ual.es (P.S.-L.)

**Keywords:** emotional intelligence, student athletes, sex

## Abstract

The concept of emotional intelligence is related to the recognition of our own emotions, their regulation and our state of mind. Additionally, it is increasingly relevant in society in general, and in the field of sport in particular. The aim of this paper is to analyze the relationship between emotional intelligence and the theory of self-determination, specifically interpersonal relationships. For this purpose, sex was taken as a mediating variable, and a structural equation model was estimated through mediation. The sample was made up of a total of 165 active sportsmen and sportswomen who are studying undergraduate and master’s degree courses related to physical activity and sport sciences. The results show that gender acts as a mediating variable between emotional intelligence and relationships with others, becoming a mediating variable of two previously unrelated variables. The implications of these results lead us to study both emotional intelligence and its importance in the field of sport, as well as the fact of paying attention to the differences that may exist in this case depending on gender.

## 1. Introduction

One of the essential purposes of sport psychology is to discover those psychological variables that are appreciable for the optimal performance of the athlete, as well as to investigate how to enhance them [1]. In this sense, emotional intelligence has been postulated as one of the most current constructs in sport psychology in the XXI century, with the largest number of investigations at present [2,3,4,5]. Emotional intelligence can be defined as the emotional response that the athlete offers inter- or intrapersonally [6], accompanied by emotional control by athletes in decision making [7] and in their own performance [8].

Within the concept of emotional intelligence itself, we bet on the ability models, specifically the established model, which is characterized by being composed of different sections or branches and the fact that each of them has a series of skills [6]:Emotional perception: This is the ability both to characterize and to explore one’s own and other people’s feelings. It involves, therefore, interest in and analysis of different signs about the expression, sensations and sincerity of emotions.Emotional understanding: This is about investigating, cataloguing and exploring emotions (both one’s own and those of others) in a retrospective way.Emotional regulation: This is about welcoming, examining and rethinking emotions in order to take advantage of them and their usefulness, both interpersonally and intrapersonally.

There are numerous investigations that have taken athletes as a sample to find out the role that emotional intelligence has in the sports field, finding that those athletes who have higher levels of emotional intelligence have greater effectiveness at the competitive level [9]. The positive correlation between emotional intelligence factors and self-concept dimensions has also been explored [10], as well as the fact that anxiety levels correlate negatively with EQ or stress [8]. Furthermore, it has been shown that in the case of motivation, there is a positive correlation with EQ [11,12,13,14]. However, there is not much agreement on the differences established between sexes in this regard [10,15,16,17].

On the other hand, the self-determination theory [18] points out the impulses that lead a subject to initiate and pursue a behavior [19]. Within this theory, we also find the importance of three basic psychological needs: competence, autonomy and relationships with others [20]. Thus, competence refers to the way in which a person relates to and copes with daily life effectively and confidently [21], while autonomy corresponds to the person’s own decisions and relatedness refers to the interpersonal relationships that the person establishes [22]. Recent studies have shown that there is a direct and positive relationship between emotional intelligence and autonomy, and emotional intelligence and competence; however, such studies have found that emotional intelligence and relatedness do not establish a positive relationship [23]. Therefore, it is worth asking whether, in the face of these facts, variables such as sex—taking into account the different scores in emotional intelligence depending on it—can become a mediating variable in the relationship between motivation and emotional intelligence in athletes. Thus, through this research work, we intend to analyze whether gender is a mediating variable between emotional intelligence (attention to feelings, emotional clarity and repair of emotions) and motivation from the point of view of the self-determination theory, specifically in the dimension referring to relationships with other people.

This research is divided into the following sections. In the introduction, the subject of emotional intelligence and the theory of self-determination is approached from a conceptual point of view. Next, the methodology section describes the main objective of this work and the method employed (indicating participants, instruments and data analysis). Next, the results of this research are presented, which show that the gender variable mediates between emotional intelligence (attention, clarity and emotional regulation) and self-determination theory (relationships with others), two variables that, a priori, did not have a direct and positive relationship. Finally, the conclusions of this work are intended to emphasize the excellence of an intervention that is not only cognitive, but also psychological and emotional, in relation to the competencies of the athlete in the context in which he/she develops.

## 2. Materials and Methods

The method used was correlational, corresponding to an ex post facto design, and retrospective and comparative in nature, since the dimensions of emotional intelligence are compared with other types of variables, in this case dependent on the relationship with others.

### 2.1. Participants

The sample consisted of 165 undergraduate and master’s degree students related to Physical Activity and Sport Sciences. The significance of the age of the sample was 20.33 years, with a standard deviation SD = 3.44. Regarding sex, 70.9% (*n* = 117) were male and 27.9% (*n* = 46) were female. The sample size was determined according to the number of students who, after information and consent, decided to participate in the study. The questionnaire was administered to four undergraduate courses in physical activity and sport sciences. The questionnaire was administered to students of a master’s degree in teaching (specializing in physical education) and a master’s degree in sports science research. All participants completed an official informed consent form of the University of Almeria (Spain) and were informed of the data protection protocol.

### 2.2. Instruments

The instruments used in this work are the following: The TMMS-24 [22], which was created to measure emotional intelligence from the model [6]. It addresses three dimensions, such as perception, understanding and regulation of emotions, through a Likert-type scale [22]. A Cronbach’s alpha = 0.84 was obtained for this work. Moreover, it showed high reliability (Cronbach’s alpha) for each dimension (perception, α = 0.90; clarity, α = 0.90; regulation α = 0.86) and adequate test–retest reliability: perception = 0.60; comprehension = 0.70 and regulation = 0.83 [24]. In this work, the TMMS-24 [9] was used as an instrument for measuring emotional intelligence, in line with the theoretical model of emotional intelligence [10]. This instrument is a self-report measure of perceived emotional intelligence; that is, the self-knowledge that a person possesses about his or her own emotional abilities: attention to feelings, emotional clarity and emotion repair.The scale of sport motivational mediators [24] assesses the satisfaction of basic psychological needs in the basic psychological needs in the field of sport on a Likert-type scale. It is composed of 23 items grouped into 3 factors: (a) autonomy: 8 items; (b) perceived competence: 7 items; (c) relationship with others: 8 items. Only those corresponding to factor C were chosen. For this work, a Cronbach’s alpha = 0.71 was obtained. The reliability of the original instrument was 0.75 for the first factor; 0.76 for the second; and 0.71 for the third [25].

### 2.3. Data Analysis

The data analyses used in this study were descriptive statistics (mean, standard deviation and bivariate correlations), reliability analysis and structural equation modeling (SEM), in contrast with the relationships established in the hypothesized model. Specifically, the mediation test was used to study whether a variable affects another variable [26], analyzing the effect of sex on the dependent variable (Table 1).

## 3. Results

The importance of conducting a mediation test lies in the intention to find out the role that a third variable—in this case, gender—plays in emotional intelligence and in relationships with others. This allows us to understand the effect of the sex variable, together with the other two variables. Therefore, the results show a simple mediation model in which gender plays the role of the mediating variable.

For this, a structural equation model has been used because of the possibility of controlling for measurement error, the ease of using multiple indicators of the constructs and the variety of measures of fit that the models provide, among other reasons. Specifically, the indirect effect test assesses the fit produced by the mediated effect model under the restricted modeling conditions, and thus, the mediation effect is tested.

In principle, although the correlation between sex and relationships is very high, it is significantly different from one, as one is not included in the confidence interval between the two variables [0.928, 0.962]. Therefore, the three variables can be in the same model, and there is no problem of linear dependence.

Furthermore, we can affirm that mediation exists, since the indirect effect is significant, as can be seen in the value of the parameter (0.753, *p*-value = 0.000). As can be seen in Figure 1, the sex variable acts as a mediator between the emotional intelligence variable. It also does so with the variable relationships with others. Therefore, the results of this study show that the sex variable is a mediator between two variables that, a priori, did not have a direct and positive relationship, such as emotional intelligence and relationships with others.

## 4. Discussion and Conclusions

The main findings of this work lie mainly in the demonstration that when the gender variable is taken into account, the variables related to emotional intelligence (attention, clarity and emotional regulation) and self-determination theory (particularly relationships with others) are shown to be mediated. That is, a third variable—gender—establishes the existence of a relationship between two previously unrelated variables separately, without this third variable.

These results are important, drawing attention to the differences that may exist in this case according to gender and EQ and gender and self-determination theory, which coincide with previous research, which, like our work, has highlighted the fact that high levels of emotional intelligence correlate with higher satisfaction of the three basic psychological needs [16,27,28,29,30,31,32,33,34].

In this way, within the field of sport, we intend to highlight the relevance of an intervention that is not only cognitive, but also psychological and emotional, in relation to the competencies of the athlete in the context in which he or she develops. We aim to do this without leaving aside the relevance of interpersonal relationships, mainly in team sports, group cohesion and the feeling of belonging to the group. All of these are essential to promote the social development, enrichment and personal well-being of the athlete.

We must not forget to be cautious with these results due to the sample size. Therefore, we consider that it would be convenient to replicate the study with larger samples in order to be able to contrast these findings, as this is one of the current limitations of this work.

Future lines of research will aim to determine whether there are differences according to the type of sport practiced. Thus, future studies will attempt to analyze differences according to the degree of professionalization of each sport practiced by future participants.

## Figures and Tables

**Figure 1 ijerph-19-09212-f001:**
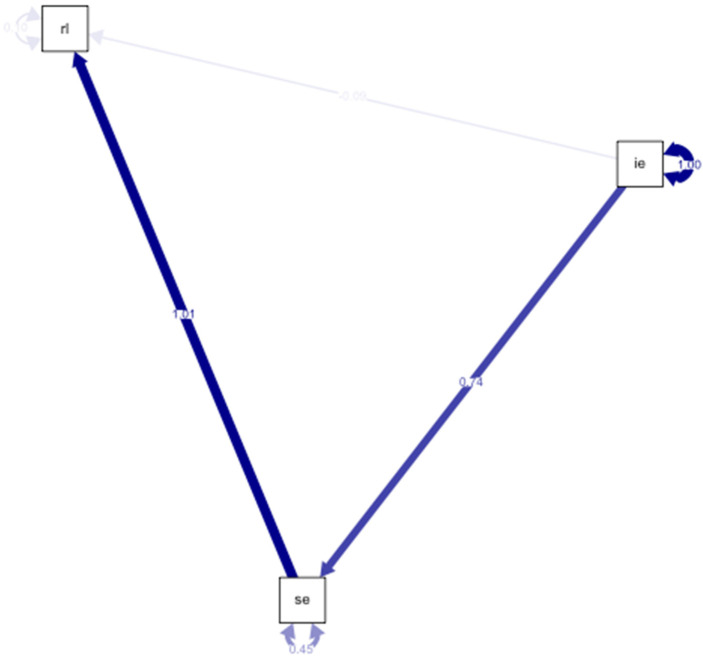
Mediation model. Note. se: sex; rl: relation; ie: emotional intelligence.

**Table 1 ijerph-19-09212-t001:** Preliminary analysis.

	1	2	3
1. Sex		0.027	−0.177 *
2. Relation			0.235 **
3. Emotional intelligence			

Note. * *p* < 0.05; ** *p* < 0.01.

## Data Availability

Not applicable.

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
