# Peer review of "Emotional Intelligence, Interpersonal Relationships and the Role of Gender in Student Athletes"

_ijerph, 2022, doi:10.3390/ijerph19159212_

Round 1

Reviewer 1 Report

The aim of this article is analysis whether there is a mediation between sex and emotional intelligence (attention to feelings, emotional clarity, and recovery of emotions) and self-determination theory (interpersonal relationships).

I note that the article needs to be improved.

In the Discussions and Conclusions section, information has appeared that must be in the Introduction.

In the Conclusion, it is necessary to present the results of the article. The question of which results are trivial or non-trivial is, of course, the choice of the editor. In the Conclusion of the study, questions are noted that clarify why it is necessary.

It is also not entirely clear why the authors write the principle "emotional intelligence" with a capital letter.

The Discussion and Conclusion section needs to be finalized as it is essentially an Introduction.

In the Results section, two of the three paragraphs begin with "In principle". It needs to be reformulated. The results look like common sense information.

There is a correlation. All. If the authors were puzzled by comparing the results in teams where such relationships exist and where they do not, this would prove something.

You can go the other way. There are more options in the Methodology section than in the conclusions. If the authors finalize the Results section according to the measurements stated in the Methodology, this will add weight to the work.

Author Response

  1. In the section of Discussions and Conclusions has appeared information that should be in the Introduction.

This section has been restructured by adding the objective of the research, the main findings, the implications it entails, the limitations found and future line of research.

  1. In the Conclusion, it is necessary to present the results of the article. The question of which results are trivial or non-trivial is, of course, the editor's choice.

This section has been restructured by adding the objective of the research, the main findings, the implications it entails, the limitations found and future line of research.

  1. It is also not entirely clear why the authors write the principle "emotional intelligence" with a capital letter.

The criterion has been established to always write both terms in lowercase

  1. The Discussion and Conclusion section needs to be finalized as it is essentially an Introduction.

This section has been restructured by adding the objective of the research, the main findings, the implications it entails, the limitations found and future line of research.

  1. In the Results section, two of the three paragraphs begin with "In principle." It needs to be reformulated.

This has been modified and the results have explained the type of test performed.

  1. The results look like common-sense information. There is a correlation. All. If the authors were puzzled to compare the results on teams where such relationships exist and where there are none, this would prove something. You can go the other way. This has been modified and the results have explained the type of test performed.
  2. If the authors finish the Results section according to the measures set out in the Methodology, this will add weight to the work.

The results have been rewritten taking into consideration what mediation is and what are the results provided by our model.

Reviewer 2 Report

The authors addressed an tropical issue "emotional intelligence" in athletes. 

To improve the quality of the paper we suggest that the following should be taken into account; 

Abstract:

The abstract lacks a concluding statement. 

Introduction

Line 36- 40. I suggest that this part best fits the Materials and Methods section 2.2 Instruments page 2 of 6., line 82.

Materials and Methods

Line 69  "............... compared with other types of variables, in this case dependent on the relationship with others". This is not very clear to explain types of variables which authors are referring to. 

Line 75 ....consent....... This study used university students and the authors are silent on Ethical clearance from the institution. We recommend that the authors should provide the ethical clearance certificate. 

Results

Page 3 of 6. Line 112. Figure 1 .............Not....... spelling error for "Note"

Line 113 - 118. This paragraph is a repetition of line 105 - 110. 

Discussion and Conclusions. 

This section lacks depth. Line 124 - 131, the discussion is a recap of introduction which does not give a an explanation on the results under discussion. 

Author Response

  1. Summary: The summary lacks a final statement.

Information has been added to the summary and the results found have been clarified (lines 13-17).

  1. Introduction Line 36-40. I suggest that this part be a better fit for the Materials and Methods section

This information has been added in the instruments section (lines 100-104).

  1. 2 Instruments, page 2 of 6, line 82. Materials and methods Line 69 "............... compared to other types of variables, in this case dependent on the relationship with others". This is not very clear to explain types of variables to which the authors refer.

That information has been deleted

  1. Line 75 .... consent....... This study used university students and the authors are silent about the ethical authorization of the institution. We recommend that authors provide the certificate of ethical authorization.

All participants completed an official informed consent form from the University of Almería (Spain) and were informed of the data protection protocol (lines 92-93)

  1. Results Page 3 of 6. Line 112. Figure 1 ............. No....... spelling error for "Note" Line 113 - 118. This paragraph is a repetition of line 105 - 110.

Fixed the error

  1. Discussions and conclusions. This section lacks depth. Lines 124 - 131, the discussion is a summary of the introduction that does not give an explanation about the results under discussion.

This section has been restructured by adding the objective of the research, the main findings, the implications it entails, the limitations found and future line of research.

Round 2

Reviewer 1 Report

This article is devoted to the development of the concept of emotional intelligence in sports. In this paper, the authors attempt to analyze the relationship between emotional intelligence and interpersonal relationships, in which sex is a mediating variable.

The authors have worked on the article but the text still needs to be improved.

First, the authors need to proofread the text for English grammar and punctuation. Secondly, the authors should use the terminology more clearly, somewhere they use sex and somewhere gender. Ambiguity must be avoided. This test claims to be scientific and must be clear and concise with one terminology.

In lines 9-10 the term "sex" is used, and in lines 172-173, the authors use the term "gender". The scientific text should be intriguing, but the neutral term "gender" should be used.

The conclusions look unconvincing. authors put links to articles where the text should be. For example, lines 187-189: "The implications of these results lead us to study both emotional intelligence and its importance in the field of sport, as previous studies have shown that high scores in emotional intelligence correlate with greater satisfaction of the three basic psychological needs ". And then there is literature. The reader does not have to go and search the texts to see which authors of the article consider the needs to be the main ones. Need to write.

A paragraph cannot consist of one sentence. And there are plenty of those in the text.

The text looks messy.

Line 124: The sentence does not start with a capital letter.

The first and second paragraph of the conclusion looks strange, where the goal is indicated and with the help of the same turnover. Need to reformulate. And goals are an introduction. The conclusion should be devoted to demonstrating the results.

In the Introduction, it is also necessary to prescribe the logic of the work, that is, what each section with a summary is devoted to. This will allow the authors to clearly formulate their idea. Further, the Abstract needs to be finalized after the authors make their wording and terminology clear.

Author Response

  1. A paragraph cannot consist of a sentence. And there are many of those in the text.

The full text has been revised and amended in response to this suggestion.

  1. The text looks messy.

The full text has been revised and amended in response to this suggestion.

  1. Line 124: The sentence does not begin with a capital letter.

Fixed the error

  1. The first and second paragraphs of the conclusion seem strange, where the objective is indicated and with the help of the same rotation. Need to reformulate.

Both paragraphs have been reformulated

  1. And goals are an introduction.

This section has been reformulated.

  1. The conclusion should be devoted to demonstrating the results.

This section has been reformulated.

  1. In the Introduction, it is also necessary to prescribe the logic of the work, that is, what each section is dedicated to with a summary. This will allow the authors to clearly formulate their idea.

Added a clarification of each section in the introduction (lines 78-87)

  1. In addition, the Abstract should be finalized after the authors clarify their wording and terminology.

The full summary (lines 7-17) has been redone.

Reviewer 2 Report

The authors incorporated all the suggestions from the reviewer. However authors should address the following;

- Page 7-8, Line 314 -5, Reference 23 is a blank. I am unsure if this has caused the references list to be numbered to 38 as the last reference yet, in the intext (manuscript references end on 37 see line 190 page 5 of 8.

Author Response

The authors incorporated all of the reviewer's suggestions. However, authors should address the following;

 - Page 7-8, Line 314 -5, Reference 23 is a blank space. I am not sure if this has caused the list of references to be numbered at 38 as the last reference so far, in the text (manuscript references end in 37; see line 190, page 5 of 8.

Fixed this error